# Mg, K-containing microparticle: A possible active principle of a culture extract produced by a microbial consortium

Toru Higashinakagawa[1,2]*, Haruhisa Kikuchi[3¤], Hidekazu Kuwayama[4]

**1** International Center for Molecular, Cellular and Immunological Research, Tokyo Women's Medical University, Tokyo, Japan, **2** EM Research Organization, Okinawa, Japan, **3** Laboratory of Natural Product Chemistry, Tohoku University, Sendai, Japan, **4** Faculty of Life and Environmental Sciences, University of Tsukuba, Tsukuba, Japan

¤ Current address: Division of Natural Medicines, Faculty of Pharmacy, Keio University, Tokyo, Japan
* toru@waseda.jp

**Data Availability Statement:** All relevant data are within the manuscript and its Supporting Information files.

## Abstract

A synthetic microbial consortium called Effective Microorganisms (EM) consists mainly of photosynthetic bacteria, lactic acid bacteria and yeast. Various effects of EM·XGOLD, a health drink produced by EM, on life cycle of *Dictyostelium discoideum* were described previously. Here, we report our attempt to identify the active principle, termed EMF, that brought about the observed effects. Throughout the purification processes, the presence of the active principle was monitored by promoted fruiting body formation. By liquid-liquid separation the activity was recovered in aqueous phase, which, after concentration, was further subjected to reverse-phase column chromatography. No activity was detected in any eluant, while almost all the activity was recovered in residual insoluble material. The application of conventional organic chemistry procedures to the residual fraction did not lead to any informative results. Acid treatment of the insoluble material produced air bubbles, suggesting it to be composed of some inorganic carbonate. Viewed under scanning electronmicroscope, the residue revealed spherical particles of $\mu$m size range. Energy Dispersive X-ray (EDX) Spectroscopy pointed to the existence, on the surface of the particles, of magnesium and, to a certain extent, of potassium. In separate experiments, acid treatment and alkali neutralization of EM·XGOLD completely wiped out the stimulatory activity of fruiting body formation. These lines of evidence indicate these Mg, K-containing microparticles to be an active principle of EM culture extract. How these particles exert their effect is currently under intensive investigation.

## Introduction

Microbiome, the world of microorganisms, has attracted more and more attention in recent years. Metagenomic approach that deals with the collection of microorganisms in its entirety has uncovered a massive number of novel genes most of which are with unknown functions [1–3]. Of particular interest is the growing number of findings that a combination of

**Funding:** This study was supported in the form of a grant awarded from the University of Tsukuba awarded to HK.

**Competing interests:** The authors have declared that no competing interests exist.

microorganisms, or a microbial consortium, exhibits novel effects that could not be expected from purecultures [4–7]. Atarashi et al. showed that a set of selected 17 Clostridia strains is essential for highest Treg induction, with its subset being less effective [5]. A similar observation with a combined 11 bacterial strains has been reported in IFNγ⁺CD8 cell induction [6]. In these reports, synergistic cooperation contributed by constituent bacteria has been postulated for their maximum inducing activity. The exact mechanism, as well as molecular properties produced by these consortia, is yet to be identified. These trends, however, have definitely aroused renewed interests in microbial consortium, either naturally occurring or synthetic, and set up a stage for a novel applied microbiology.

EM is a short form of Effective Microorganisms and a synthetic microbial consortium which consists mainly of photosynthetic bacteria, lactic acid bacteria and yeast. EM is also a registered trademark and a brand name owned by EMRO (EM Research Organization). Since its serendipitous discovery by T. Higa in 1970s [8], EM itself or its culture extract has proved to be beneficial in numerous fields such as agriculture, bioremediation, environmental cleanup and so forth [9–12]. By functional genomics and metabolome analyses, ECEM, one of EM products, has exhibited anti-inflammatory and immunostimulatory effects in mouse macrophage [13]. Furthermore, EM·XGOLD (abbreviated as EMXG in this communication), another EM product brought to the market as a health drink, has been reported to be effective in sustaining human immunity function [14]. However, the molecular mechanism of those effects is still unknown.

Prompted by these reports and out of our genuine curiosity, we initiated a more in-depth examination of the effect of EM products on cellular activity with the use of cellular slime mold *Dictyostelium discoideum* as a model. We described in the foregoing paper various effects of EMXG on life cycle of this organism, including a novel finding, i.e. modulation of cAMP oscillation rhythm [15]. Here, we report our attempt to identify the active principle that brings about the observed effects. Several lines of evidence point to the fact that the activity resides in Mg, K-containing microparticle of several micrometer range. How these particles exert their effect is currently under intensive investigation.

## Results

### The active principle is water-soluble

ECEM was first subjected to liquid-liquid partition to see whether the activity is organic solvent-soluble or water-soluble. After extraction by ethyl acetate followed by n-butanol, the combined organic solvent soluble fraction and water-soluble fraction were assayed as described in Materials and Methods. As shown in Fig 1, most of the activity was recovered in water-soluble fraction. The lower activity observed in both organic solvent and the water soluble fractions was due to the inhibitory effect of dimethyl sulfoxide used for dissolving powder state of each fraction into water.

### The active principle was not found in the column eluates

Water-soluble fraction was then applied onto HPLC column and was eluted with a mixture of water-methanol (1:4) to give five fractions (S2 Fig). Each eluted fraction was subjected to activity assay, and none of the fractions showed the promoted fruiting body formation (Fig 2). Fraction 1 showed even an inhibitory activity. Here, possibility exists that the activity is separated into more than two fractions during the course of chromatography. To address this possibility, various combinations of eluted fraction were tested for activity assay. As seen from Fig 3, none of the combinations exhibited the promoted fruiting body formation, thus excluding the supposed possibility.

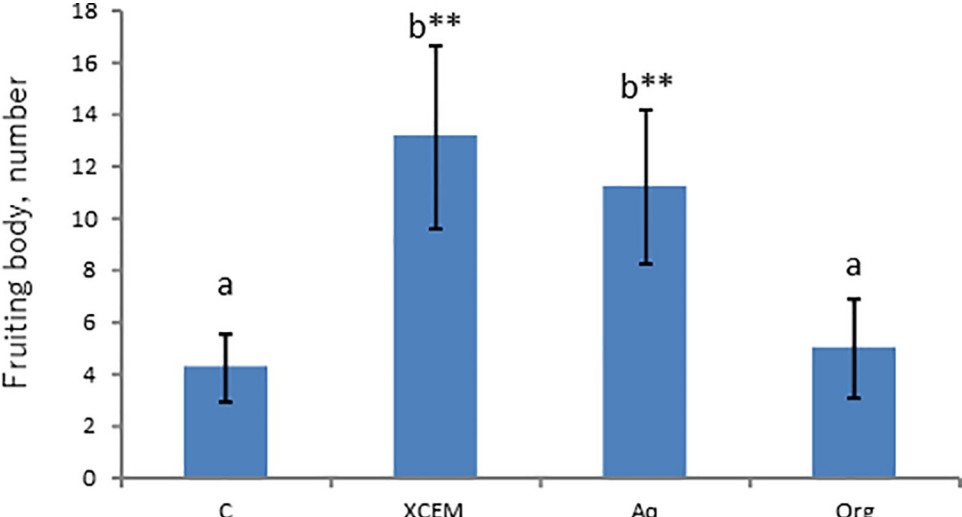

**Fig 1. Liquid-liquid partition of EMF activity.** The activity was recovered in aqueous phase. C (Control without ECEM); ECEM (10% ECEM); Aq (Aqueous phase); Org (Organic phase). **: Significant difference in comparison to the Control by Steel test ($p < 0.01$), a,b: different letters show significant difference by Steel-Dwass test($p < 0.01$).

## The active principle was recovered in the insoluble residue

S2 Fig shows that water soluble fraction, in which the activity was associated, was divided into 5 column eluants and insoluble residue. Since no activity was found in any eluants and in any of their combinations, logical possibility suggests that the activity should be found in insoluble residue, if activity is not lost during the course of chromatographic separation. We addressed this possibility. The insoluble residue, however, was not solubilized, as its name stands, by such procedures as vortexing and sonication with occasional addition of dimethyl sulfoxide. As a final trial, insoluble residue suspension equivalent to 10% ECEM was mixed with agar and water, and autoclaved according to the conventional procedure for preparing agar plates.

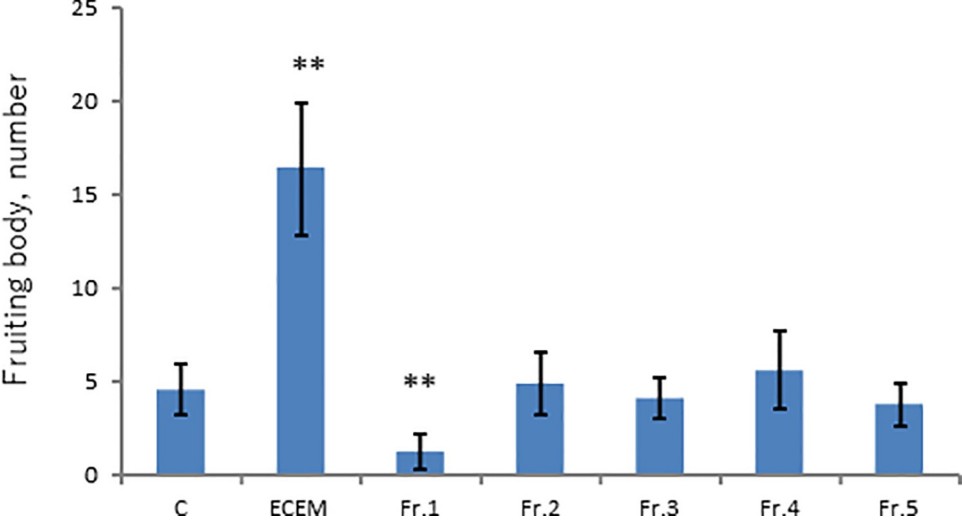

**Fig 2. Chromatographic separation of aqueous phase in Fig 1.** Aqueous phase was separated into 5 fractions and insoluble residue (S2 Fig). C (Control without ECEM); ECEM (10% ECEM). **: Significant difference in comparison to the Control by Steel test ($p < 0.01$).

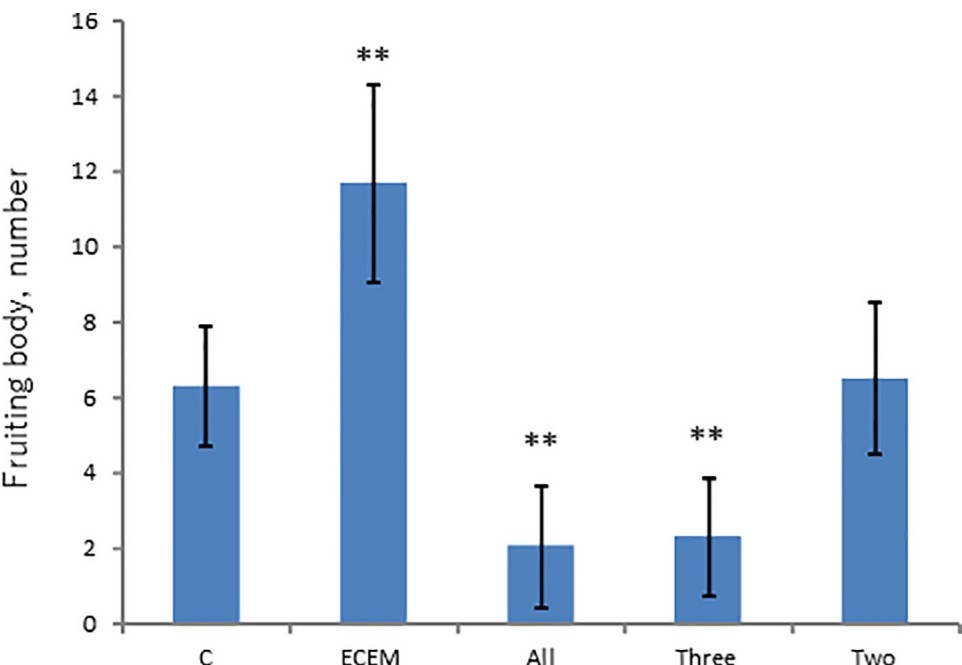

**Fig 3. Assay of combinatorial mixture of fractions in Fig 2.** C (Control without ECEM); ECEM (10% ECEM); All (All fractions combined); Three (Fr. 1, Fr. 2 and Fr. 4 are combined); Two (Fr. 3 and Fr. 5 are combined). **: Significant difference in comparison to the Control by Steel test (p<0.01).

To our little surprise, agar plate finally obtained looked smooth without showing any blobs anticipated from insoluble residue. This enabled us to proceed further to the assay procedure. Fig 4 describes the result of activity assay experiment. As seen, it is clear that almost all activity was recovered in the insoluble residue. The result that the activity is associated with insoluble residue raises several possibilities as to the nature of activity in the residue. Whatever the mode

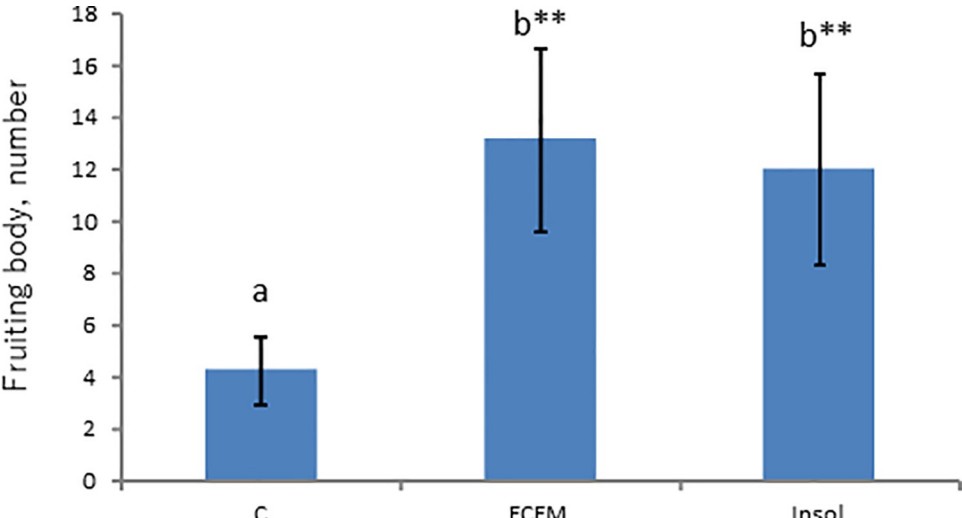

**Fig 4. Activity of fruiting body formation was recovered in the residual fraction.** C (Control without ECEM); ECEM (10% ECEM); Insol (Insoluble residue). **: Significant difference in comparison to the Control by Steel test (p<0.01), a, b: different letters show significant difference by Steel-Dwass test(p<0.01).

of existence, various organic chemistry procedures were applied, such as esterification, acid hydrolysis for detection of polysaccharide and NMR. Nothing informative could be obtained by these trials. Finally, it was found that the residue dissolves in acidified methanol with concomitant formation of bubbles, suggesting the possibility of carbonate salt, the result rather resisting rational understanding.

### Views of insoluble residue under scanning electron microscope

The insoluble residue was viewed under scanning electron microscope. Fig 5 represents a typical image which reveals spherical particles with various sizes of μm range. Application of Energy Dispersive X-ray (EDX) Spectroscopy pointed to the existence, on the surface of the particles, of magnesium and, to a certain extent, of potassium (S3 Fig). One additional observation was the matrix-like structure revealed when the residue was autoclaved, the meaning of which remains an enigma (S4 Fig). Over all, the insoluble residue consists of spherical microparticles containing mainly magnesium and, to a certain extent, potassium. So, from now on, just for convenience, we call it Mg-particle.

### Acid treatment of EMXG

In order to unarguably assign the active principle to Mg-particle, it would be the best to isolate Mg-particle in a pure form, followed by activity assay. Various procedures were attempted, such as sedimentation and equilibrium centrifugation, to no avail. As described above, it was observed that the insoluble residue dissolves into acid with concomitant release of gas. This observation suggests that one of the major constituents of the insoluble residue is $MgCO_3$, the inference apparently arguing against the reasonable understanding that the inorganic compound could hardly modulate the cellular function. If, however, the Mg-particle in the residue

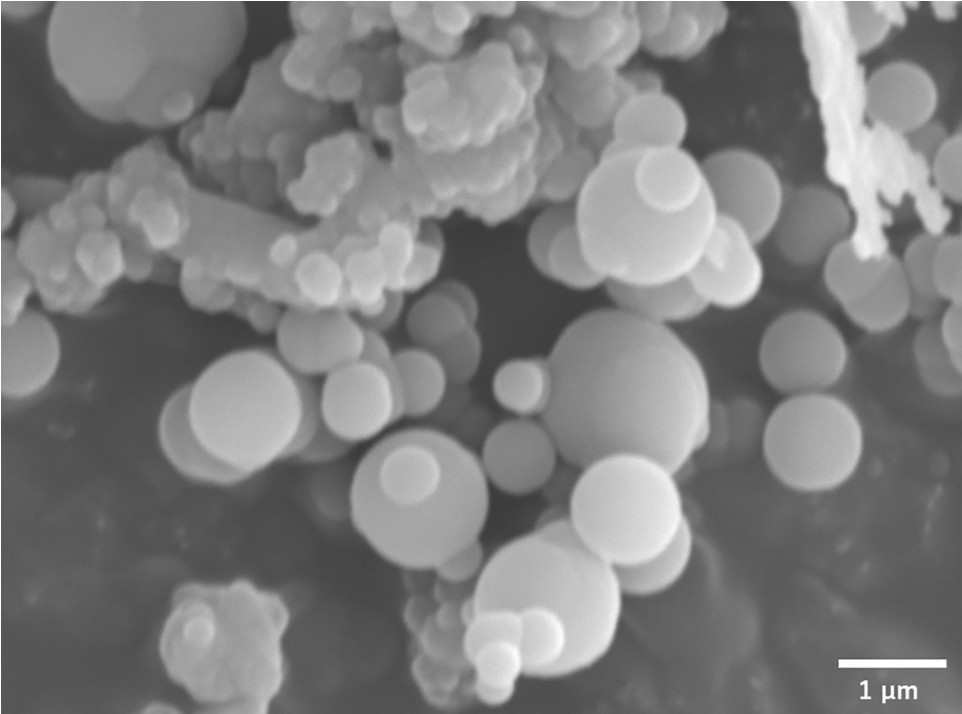

**Fig 5. Scanning electron micrograph of insoluble residue.**

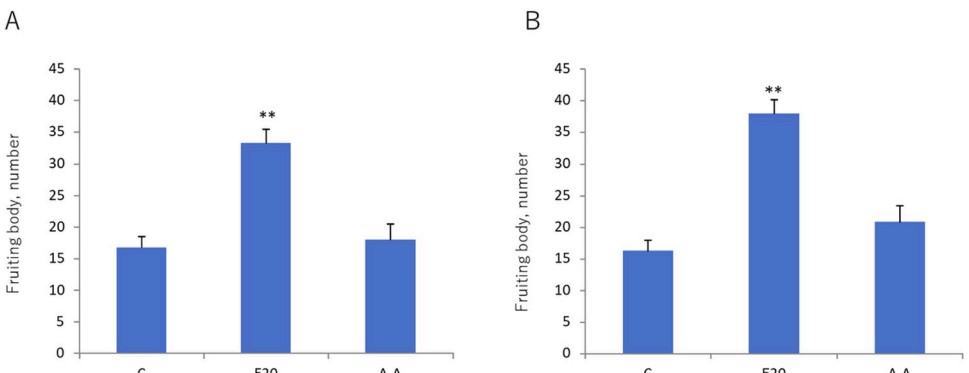

**Fig 6. EMXG was acidified with HCl followed by neutralization with NaOH.** The treated EMXG was assayed for fruiting body formation. A; Assay was performed right after acid-alkali treatment. B; The treated sample was kept at room temperature for 2 months and assayed for fruiting body formation. C (Control without ECEM); E20 (EMXG 20%); A-A (Acid treatment followed by alkali neutralization), **: Significant difference in comparison to the Control by Steel test (p<0.01).

is effective in some way to promote fruiting body formation, the acid treatment would be expected to wipe away the activity, since the salt composed by weak acid will be destroyed by strong acid. We challenged this possibility. For this experiment, we used EMXG, a diluted form of ECEM, as a test material, since acidification of ECEM resulted in the formation of insoluble precipitate and made the experiment complicated. Thus, EMXG, into which pH electrode was inserted, was added diluted HCl until pH reached 1~2. Then, with the same set up, diluted NaOH was added dropwise till pH went up to 7~8. The neutralization step was needed because the agar, which was used for activity assay, was unable to solidify under acidic condition. Fig 6A shows what resulted in one of such experiments. It is clear that the activity of fruiting body formation is almost completely wiped away by acid treatment followed by alkali neutralization. To exclude the possible effect of sodium and chloride ions produced by these treatments, an equivalent amount of NaCl was added to the control, which revealed no effect on fruiting body formation. Furthermore, to ensure that no reformation of Mg-particle or any kind of activity reversal was occurred during a long period after neutralization, the same reaction mixture was assayed 2 months after the neutralization reaction. The result is shown in Fig 6B, which is similar to that in Fig 6A. Whatever the mode of action may be, the results of this experiment suggest that Mg-particle to be a most possible candidate for the active principle of EMXG.

## Discussion

In the preceding paper, we described how EMXG affects the life cycle of *Dictyostelium discoideum* [15]. EMXG promoted the proliferation of amoeba cells, stimulated the differentiation process by formulating more of fruiting bodies and modulated the oscillation of cAMP with concomitant increase of cAMP level of the medium. Nothing is known, however, whether these multiple effects are caused by a single factor or by several different factors contained present in EMXG. As described, the follow-up procedure of active principle was through monitoring the accelerated fruiting body formation of *Dictyostelium discoideum*. Therefore, it would be appropriate to state that Mg-particles identified in this communication represent one of the active principles residing in EMXG and other EM-related products. Therefore, possibility cannot be ruled out that other factors active principles that are as distinct from Mg-particles is at work, exhibiting beneficial effects in a wide variety of fields. These aspects are yet to

be subjected to further in-depth studies, the presence of other factors being clearly in the realm of possibility.

The result of acid treatment suggests the chemical form of Mg-particle to be the carbonate form of magnesium, most probably $MgCO_3$. Based on the experiments described above, we propose the observed Mg-particle to be a most possible candidate for the active principles of ECEM and EM·XGOLD. These lines of evidence prompted us to see whether authentic $MgCO_3$ itself possesses the acceleration activity of fruiting body formation similar to the one observed with EMXG. Ten batches of $MgCO_3$ with various degree of purity and impurities were supplied by a chemical company. Ten mg or so of each were included in agar plate for the assay. It was found that only one batch of $MgCO_3$ exhibited remarkable stimulatory activity for fruiting body formation. How this particular one has the observed activity is at present in an enigma, research currently being carried out in regard to its specific crystal structure. In the current study, we did not focus on the role of potassium which was also associated with Mg-particle.

It is of great interest how inorganic materials exert any effect on biological cells. There were several preceding findings regarding the biological effect by inorganic materials. As a pioneering observation, Glass and Kennett reported that carbon material added to an agar plate or a liquid culture medium showed growth-enhancing effect on certain bacteria [16]. The plausible hypothesis for these observations was that these carbon materials act indirectly by adsorbing growth inhibitory substances from the medium. Following this observation, Matsuhashi *et al.* reported that carbon material such as graphite and activated charcoal, but not diamond, stimulated the growth of special bacteria, carbophilic strain of *Bacillus* [17]. Also, Matsuhashi *et al.* presented several lines of evidence which support the direct effect of carbon materials on living organisms [18]. They showed, through ways of effect-blocking experiments, that the effect is physical, not chemical [18]. More interestingly, the growth-enhancing signal was shown to propagate through distance, by way of the sound, termed by them biosonics [19]. Another report concerning the effect of inorganic material deals with carbon micro coil, abbreviated as carbon micro coil (CMC) [20]. CMC was produced by metal-catalyzed pyrolysis of acetylene at 700~800˚C [20]. Ogawa found that CMCs have activation effect for skin cells and their collagen mRNA production [21]. For example, the addition of CMCs promoted the growth of Pam212 mouse keratinocyte, in dose dependent fashion, upto 1.6 times versus the control for the culture period of 6 days [21]. Also, CMC was shown to suppress the growth of HeLa cells by almost 80%. Interestingly, this effect was essentially wiped out when the coiled structure was sufficiently destroyed [22]. Furthermore, Komura reported that CMC, coupled with ultrasound exposure, inhibited the growth of human hepatocarcinoma and mouse sarcoma cells, and that this effect was through generation of hydroxyl radical (•OH) in the culture medium. Komura suggested that CMC is a potent sonosensitizer and applicable to cancer therapy in combination with ultrasound [23]. At the present time, it is rather hard to make up a unified theory regarding the possible mechanisms for the observations described. It is, however, of interest to note that the steric structure might be involved based on two independent observations. Matsuhashi *et al.* showed that the effect was not detected with diamond while graphite and activated charcoal was effective [17]. In the case of CMC, the effect disappeared when the coiled structure was destroyed [22]. Our observation that some authentic $MgCO_3$ showed the stimulatory effect in fruiting body formation may have some relevance to this kind of inference.

The present communication described Mg-microparticle to be one of the active principles that exert the stimulatory effect on the growth, differentiation and cAMP oscillation rhythm in cellular slime mold, *Dictyostelium discoideum*. How this effect is exerted, i.e. what kind of molecules stands between Mg-particle and fruiting body formation prompts our further interest, and is under intensive investigation.

## Materials and methods

### EM, ECEM and EM·XGOLD

EM represents a short form of Effective Microorganisms and is a registered trademark and a brand name owned by EMRO (EM Research Organization). EM is a synthetic microbial consortium consisting mainly of photosynthetic bacteria, lactic acid bacteria and yeast. ECEM is an extract from cultured EM. EM·XGOLD (Abbreviated as EMXG in this communication) is a diluted form of ECEM and brought to the market as a health drink [14].

### Assay for the active principle

During purification processes, active principle was monitored by the promoted fruiting body formation of *Dictyostelium discoideum* [15; S1 Fig].

### Liquid-liquid partition

ECEM was extracted with ethyl acetate three times. The ethyl acetate layer was concentrated *in vacuo* to yield ethyl acetate soluble fraction. The water layer was extracted with n-butanol three times. The n-butanol and water layers were concentrated *in vacuo*, respectively, to give n-butanol soluble and water-soluble fractions. The ethyl acetate soluble and n-butanol soluble fractions were combined as organic solvent soluble fraction.

### Column chromatography

Two hundred g of dried water-soluble fraction were dissolved in 10 mL of water-methanol (1:4) mixture, and the solution was filtered through filter paper to give the filtrate and the insoluble fraction (31.5 mg). The filtrate was subjected to preparative HPLC. The column was eluted with a mixture of water-methanol (1:4) to give five fractions (S2 Fig).

### Electrommicroscopy

Insoluble residual fraction was resuspended in water, and a drop of it was placed onto a carbon tape attached to a sample stage, air fried and further subjected to gold deposition. The specimen was viewed with scanning electron microscope, LA101, JOEL. The atomic element was analyzed with Energy Dispersive X-ray (EDX) Spectroscopy equipment.

### Acid treatment

EMXG was acidified by dropwise addition of HCl until its pH reaches around 2. Then, it was neutralized by addition of NaOH until its pH reaches 7 ~ 8.

## Supporting information

**S1 Fig. Assay for the active principle.** Twenty droplets of AX2 amoeba cell suspension, containing ca. 8,000 cells, were spotted onto non-nutrient agar plate which contained a separated fraction to be assayed. Forty-eight hours later, number of fruiting body was counted under dissecting microscope, and averaged to give a number for one spot. Control was composed of agar and water. Positive control contained 10% ECEM. For monitoring the active principle during purification processes, agar plate contained the separated fraction in an amount equivalent to 10% ECEM.
(TIF)

**S2 Fig. Column chromatography of water-soluble fraction.** Two hundred g of dried water-soluble fraction were dissolved in 10 mL of water-methanol (1:4) mixture, and the solution was filtered through filter paper to give the filtrate and the insoluble (31.5 mg). The filtrate was subjected to preparative HPLC (column, JAIGEL-GS310 (20 mm x 500 mm, Japan Analytical Industry, Co. Ltd.). The column was eluted with a mixture of water-methanol (1:4) to give five fractions.
(TIF)

**S3 Fig. Energy Dispersive X-ray (EDX) spectroscopy.** The downward arrows point to signals of Mg and K. Other signals are from the background.
(TIF)

**S4 Fig. Scanning electron micrograph of insoluble residue after autoclave treatment.** Note the exposed mesh-like structures.
(TIF)

**S1 File.**
(XLSX)

## Acknowledgments

We thank Toshiko Koizumi for electronmicroscpic analysis, Shuichi Okumoto for statistical analysis and Konoshima Chemical Co., Ltd. (Osaka, Japan) for a kind supply of various batches of MgCO$_3$.

## Author Contributions

**Investigation:** Toru Higashinakagawa, Haruhisa Kikuchi, Hidekazu Kuwayama.

**Writing – original draft:** Toru Higashinakagawa.

**Writing – review & editing:** Hidekazu Kuwayama.

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
