## [Decision Letter · Decision Letter 0]

30 Jun 2021

PONE-D-21-17352

Mg, K-containing microparticle: a possible active principle of a culture extract produced by a microbial consortium

PLOS ONE

Dear Dr. higashinakagawa,

Thank you for submitting your manuscript to PLOS ONE. After careful consideration, we feel that it has merit but does not fully meet PLOS ONE’s publication criteria as it currently stands. Therefore, we invite you to submit a revised version of the manuscript that addresses the points raised during the review process.

We look forward to receiving your revised manuscript.

Kind regards,

Stefan Wölfl, Ph.D.

Academic Editor

PLOS ONE

Additional Editor Comments (if provided):

The manuscript in its present form is not yet suited and needs some minor improvments, that are substantial, before the manuscript can be accepted for publication.

There are several point mentioned and addressed by the reviewers that need to be corrected in a revised version. I also recommend careful language editing before resubmission.

For details on the required changes please carefully address all points raised be the reviewers. Because it may not be clear from the comment from reviewer one, I like to add that the authors should add and discusse more carefully the state of the art regarding possible mechanisms.

Journal Requirements:

"NO". 

6. We note that Figure(s) 5 in your submission contain copyrighted images. All PLOS content is published under the Creative Commons Attribution License (CC BY 4.0), which means that the manuscript, images, and Supporting Information files will be freely available online, and any third party is permitted to access, download, copy, distribute, and use these materials in any way, even commercially, with proper attribution. For more information, see our copyright guidelines: http://journals.plos.org/plosone/s/licenses-and-copyright.

a. You may seek permission from the original copyright holder of Figure(s) 5 to publish the content specifically under the CC BY 4.0 license. 

Reviewers' comments:

Reviewer's Responses to Questions

**Comments to the Author**

1. Is the manuscript technically sound, and do the data support the conclusions?

Reviewer #1: Yes

Reviewer #2: Yes

2. Has the statistical analysis been performed appropriately and rigorously? 

Reviewer #1: Yes

Reviewer #2: Yes

3. Have the authors made all data underlying the findings in their manuscript fully available?

Reviewer #1: Yes

Reviewer #2: Yes

4. Is the manuscript presented in an intelligible fashion and written in standard English?

Reviewer #1: Yes

Reviewer #2: Yes

5. Review Comments to the Author

Reviewer #1: The work Higashinakagawa et al. describes stimulatory effects of an extract from culture of Effective Microorganisms (ECEM) and a health drink containing ECEM (EM XGOLD) on cellular activity of model organism Dictyostelium discoideum. Higashinakagawa et al. show that inorganic material such as magnesium and potassium containing microparticles can be active substances of ECEM. However, no statement about possible mechanisms of this effect is included. Authors only discuss few papers regarding the biological effect of inorganic materials and write that this subject is currently under their investgation.

This simple manuscript is clearly written. The methodological data are well documented by nice micrographs, graphs and by statistic evaluation and I do not have not any major reservations. I believe this work will interest many workers in the field of cell biology and action of inorganic compounds on cellular functions as well as in nutrition supplements.

Reviewer #2: Higashinakagawa et al, Mg, K-containing microparticle: a possible active principle of a culture extract produced by a microbial consortium.

Higashinakagawa et al reported a very interesting and intriguing study and explain how simple elements have shown significant biological activity, namely, as they called the Effective Microorganisms (EM) on life cycle of Dictyostelium discoideum. They carried out systematic and careful studies of such EM･XGOLD, which is a well-known and popular health drink in Japan. This study is a continuation from their report in 2019 that “The life cycle of Dictyostelium discoideum is accelerated via MAP kinase cascade by a culture extract produced by a synthetic microbial consortium.”

They used various methods including i) Liquid-liquid partition, ii) column chromatography, iii) electron microscopy, iv) elemental analysis using Energy Dispersive X-ray (EDX) Spectroscopy, v) acid treatment and alkali pH titrations to figure out the major component of the biological activity. Through their careful observations and detailed studies, they conclude that the Mg, K-containing microparticle biological effect could be due to MgCO3. Although intriguing, their careful data and systematic analysis justify their conclusion. They may have solved one of long-standing puzzles in the popular health drink in Japan.

Some minor points:

1) They use the term EM for Effective Microorganisms. This abbreviation is not necessary since there are only 2 words. EM commonly refers to Electron Microscopy. It is rather confusion to read EM but it refers to something totally different. The legendary Francis Crick advised this reviewer 30 years ago to avoid use all unnecessary abbreviations and acronyms, so the readers can understand the articles better and avoid confusions.

2) Please write out ECEM on line 83, page 5.

3) Please write out CMC (carbon microcoils) on line 238, page 9. CMC commonly refers to critical micelle concentration.

4) Please add more figure legend to Figure 5, provide more information about the microparticle, why the sizes cross a wide range. Do they all have similar elemental contents?

5) On lines 183-184, they wrote “The result was exactly as shown in Fig. 6”. Please replace the word exactly with “similar” since the results in Figure are not exactly, but similar. For example, in Panel A, E20 is ~33% and in Panel B, the E20 is ~38%. They are similar but not exactly identical.

6) On line 221, page 9, they wrote “we have no idea regarding the functional role and the state of being of potassium……”. Please change to “in the current study, we did not focus on the study of potassium”.

After they make these corrections and changes, this reviewer highly recommends publication in PLoS ONE.

6. PLOS authors have the option to publish the peer review history of their article (what does this mean?). If published, this will include your full peer review and any attached files.

Reviewer #1: No

Reviewer #2: **Yes: **Shuguang Zhang

---

## [Author Response · Author response to Decision Letter 0]

22 Jul 2021

Academic Editor:

Comment: There are several point mentioned and addressed by the reviewers that need to be corrected in a revised version. I also recommend careful language editing before resubmission.

Reply: Yes, we did it.

Comment: For details on the required changes please carefully address all points raised be the reviewers. Because it may not be clear from the comment from reviewer one, I like to add that the authors should add and discusse more carefully the state of the art regarding possible mechanisms.

Reply: In the revised manuscript, we discussed to some extent the state of art regarding the possible mechanisms. 

Journal Requirements:

Comment: Thank you for stating the following financial disclosure: 

Reply: We stated the financial disclosure ‘This study was supported in the form of a grant awarded from the University of Tsukuba to HK.’

Comment: At this time, please address the following queries:

Reply: Our study was funded from a grant awarded from the University of Tsukuba to HK.

Comment: State what role the funders took in the study. If the funders had no role in your study, please state: “The funders had no role in study design, data collection and analysis, decision to publish, or preparation of the manuscript.”

Reply: The funders had no role in study design, data collection and analysis, decision to publish, or preparation of the manuscript.

Comment: If any authors received a salary from any of your funders, please state which authors and which funders.

Reply: We did not receive any salary from any funders.

Comment: If you did not receive any funding for this study, please state: “The authors received no specific funding for this work.”

Reply: We got funding from a grant awarded from the University of Tsukuba to HK.

Comment: Please include your amended statements within your cover letter; we will change the online submission form on your behalf.

Reply: We did so as you suggested.

Comment: We note that you have included the phrase “data not shown” in your manuscript. Unfortunately, this does not meet our data sharing requirements. PLOS does not permit references to inaccessible data. We require that authors provide all relevant data within the paper, Supporting Information files, or in an acceptable, public repository. Please add a citation to support this phrase or upload the data that corresponds with these findings to a stable repository (such as Figshare or Dryad) and provide and URLs, DOIs, or accession numbers that may be used to access these data. Or, if the data are not a core part of the research being presented in your study, we ask that you remove the phrase that refers to these data.

Reply: We removed the sentence which includes “data not shown”.

Comment: We note that Figure(s) 5 in your submission contain copyrighted images. All PLOS content is published under the Creative Commons Attribution License (CC BY 4.0), which means that the manuscript, images, and Supporting Information files will be freely available online, and any third party is permitted to access, download, copy, distribute, and use these materials in any way, even commercially, with proper attribution. For more information, see our copyright guidelines: http://journals.plos.org/plosone/s/licenses-and-copyright.

Reply: Fig.5 and Supplementary Fig. 4 in the submitted manuscript are the raw photographic data which we obtained directly from the scanning electron microscope. However, in order not to invite unnecessary confusion or misunderstanding, we removed the bottom end of the photograph and inserted a scale bar of 1 μm to Fig.5 and Supplementary Fig. 4.

Comment: Please include captions for your Supporting Information files at the end of your manuscript, and update any in-text citations to match accordingly. Please see our Supporting Information guidelines for more information: http://journals.plos.org/plosone/s/supporting-information.

Reply: Done as requested.

Reviewers' comments:

Reviewer's Responses to Questions

Reviewer #2:

Comment 1: They use the term EM for Effective Microorganisms. This abbreviation is not necessary since there are only 2 words. EM commonly refers to Electron Microscopy. It is rather confusion to read EM but it refers to something totally different. The legendary Francis Crick advised this reviewer 30 years ago to avoid use all unnecessary abbreviations and acronyms, so the readers can understand the articles better and avoid confusions.

Reply 1: EM is not simply an abbreviation. EM is a registered trademark and a brand name owned by EMRO (EM Research Organization). EM refers to a synthetic microbial consortium and is used over 100 countries. We already used this word, EM, in our previous paper (Ref.15 of this manuscript). On visiting web site emrojapan.com, more information about EM is available. Accordingly, we changed line 263 -265 as follows.

“EM represents a short form of Effective Microorganisms and is a registered trademark and a brand name owned by EMRO (EM Research Organization). EM consists mainly of photosynthetic bacteria, lactic acid bacteria and yeast.”

Comment 2: Please write out ECEM on line 83, page 5.

Reply 2: In compliance with the comment of Reviewer #2, we left out the term ECEM.

Comment 3: Please write out CMC (carbon microcoils) on line 238, page 9. CMC commonly refers to critical micelle concentration.

Reply 3: The term CMC for Carbon Micro Coil is an abbreviation made not by us, but was coined by its inventor, S. Motojima. Ref. 20 of this manuscript shows how it was termed. Accordingly, we changed line 238 (236 in the revised manuscript) as follows.

“carbon micro coil (CMC)[20].”

Comment 4: Please add more figure legend to Figure 5, provide more information about the microparticle, why the sizes cross a wide range. Do they all have similar elemental contents?

Reply 4: In compliance with the suggestion by Reviewer #2, we provided more information about the microparticle in the figure legend to Fig.5.

Comment 5: On lines 183-184, they wrote “The result was exactly as shown in Fig. 6”. Please replace the word exactly with “similar” since the results in Figure are not exactly, but similar. For example, in Panel A, E20 is ~33% and in Panel B, the E20 is ~38%. They are similar but not exactly identical.

Reply 5: In compliance with the suggestions by Reviewer #2, we replaced the word “exactly” with “similar” in the revised manuscript.

Comment 6: On line 221, page 9, they wrote “we have no idea regarding the functional role and the state of being of potassium……”. Please change to “in the current study, we did not focus on the study of potassium”.

Reply 6: In compliance with the suggestion by Reviewer #2, we changed the wording in the revised manuscript.

Question: PLOS authors have the option to publish the peer review history of their article (what does this mean?). If published, this will include your full peer review and any attached files.

Reply: No

---

## [Decision Letter · Decision Letter 1]

6 Oct 2021

PONE-D-21-17352R1Mg, K-containing microparticle: a possible active principle of a culture extract produced by a microbial consortiumPLOS ONE

Dear Dr. higashinakagawa,

Thank you for submitting your manuscript to PLOS ONE. After careful consideration, we feel that it has merit but does not fully meet PLOS ONE’s publication criteria as it currently stands. Therefore, we invite you to submit a revised version of the manuscript that addresses the points raised during the review process.

We look forward to receiving your revised manuscript.

Kind regards,

Stefan Wölfl, Ph.D.

Academic Editor

PLOS ONE

Journal Requirements:

Additional Editor Comments (if provided):

Thank you for the submission of the revised version. As the reviewers point out most of their major concerns have been addressed in the revision. There is however still one aspect that I do not fully understand at that makes the manuscript difficult to understand and unclear regarding the Intension of the work. The work is solely used on the trademark registered EM consortium of microorganisms from EM research organization. This is a commercial product and registered trademark. This needs to be clear in the manuscript and it will be necessary to stay potential conflicts of interest regarding the impact the work will have on the commercialization of EM XGold or EM(r).

Reviewers' comments:

Reviewer's Responses to Questions

**Comments to the Author**

1. If the authors have adequately addressed your comments raised in a previous round of review and you feel that this manuscript is now acceptable for publication, you may indicate that here to bypass the “Comments to the Author” section, enter your conflict of interest statement in the “Confidential to Editor” section, and submit your "Accept" recommendation.

Reviewer #1: All comments have been addressed

Reviewer #2: All comments have been addressed

2. Is the manuscript technically sound, and do the data support the conclusions?

Reviewer #1: Yes

Reviewer #2: Yes

3. Has the statistical analysis been performed appropriately and rigorously? 

Reviewer #1: N/A

Reviewer #2: Yes

4. Have the authors made all data underlying the findings in their manuscript fully available?

Reviewer #1: Yes

Reviewer #2: Yes

5. Is the manuscript presented in an intelligible fashion and written in standard English?

Reviewer #1: Yes

Reviewer #2: Yes

6. Review Comments to the Author

Reviewer #1: (No Response)

Reviewer #2: The authors have addressed the reviewer's concerns and suggestions. They have made corrections and revised the manuscript. This reviewer now recommends publication of this very interesting paper.

7. PLOS authors have the option to publish the peer review history of their article (what does this mean?). If published, this will include your full peer review and any attached files.

Reviewer #1: No

Reviewer #2: No

---

## [Author Response · Author response to Decision Letter 1]

15 Oct 2021

October 15, 2021

PLoS One

Dear Dr. Stefan Wölfl:

Please find herewith our revised manuscript entitled “Mg, K-containing microparticle: a possible active principle of a culture extract produced by a microbial consortium” by Toru Higashinakagawa, Haruhisa Kikuchi and Hidekazu Kuwayama.

In compliance with the Editor’s suggestion, we stated in Introduction that EM is a registered trade mark and a brand name owned by EMRO (EM Research Organization). We stated that this work was initiated based on genuine scientific curiosity, since many papers on EM thus far published described only phenomenological observations. We were truly curious to know how EM works in a more simple system like cellular slime mold. This point was stated in the third paragraph of the introduction in the revised manuscript.

The Editor also asked about the intension of our work. To answer this question, I will describe how this work started and then proceeded. Almost 10 years ago, Kuwayama, the 3rd author of the present paper, expressed his interest in EM•XGOLD which was/is freely available on the market. Kuwayama worked on cellular slime mold Dictyostelium discoideum for many years, and had been looking for any biological substances which may affect the life cycle of Dictyostelium discoideum. He had heard the rumor about EM•XGOLD and asked me to collaborate with him for looking at the effect of EM•XGOLD on the life cycle of this slime mold. I was then just retired from Waseda University, Tokyo, but still got a lab space at Tokyo Women’ Medical University. Thus, our joint work started in 2010. Since my place at TWMU did not have any facilities to grow Dictyostelium discoideum, most experiments were carried out in Kuwayama’s lab, 3 hours’ train ride away. In 2019, our first paper along this line was accepted in Journal of Molecular Microbiology and Biotechnology (JMMB, 29, 35-42, 2019). The JMMB paper described the differentiation promoting activity, namely promotion of fruiting body formation, in Dictyostelium discoideum, among other interesting effects. We found those observations quite interesting and proceeded to identify what the active principle of EM•XGOLD really is. At this point, we asked Kikuchi, the 2nd author of the present manuscript, to join our group. Kikuchi was/is an organic chemist and contributed to the identification of biologically active substances from a variety of bio-organisms including Dictyostelium discoideum. Thus, our joint work started for identifying the active principle of EM•XGOLD which lead to the present manuscript.

As described above, our work is genuinely based on the scientific curiosity. We have never intended to advertise EM•XGOLD to the society. Moreover, our result refers only to cellular slime mold Dictyostelium discoideum, never meaning to address humans. Accordingly, we do not think the information inherent in this manuscript would not affect the market situation.

We hope the present manuscript will meet the publication in PLOS One.

Thank you for your consideration. I look forward to hearing from you.

Sincerely,

Dr. Toru Higashinakagawa

International Center for Molecular, Cellular and Immunological Research

Tokyo Women’s Medical University

8-1 Kawada-cho, Shinjuku

Tokyo 162-8666, Japan

TEL: +81-3-3353-8112 ( ext. 24062 )

FAX: +81-3-3352-3088

E-mail: toru@waseda.jp

---

## [Editor Report · Decision Letter 2]

18 Oct 2021

Mg, K-containing microparticle: a possible active principle of a culture extract produced by a microbial consortium

PONE-D-21-17352R2

Dear Dr. higashinakagawa,

We’re pleased to inform you that your manuscript has been judged scientifically suitable for publication and will be formally accepted for publication once it meets all outstanding technical requirements.

Kind regards,

Stefan Wölfl, Ph.D.

Academic Editor

PLOS ONE
---

## [Editor Report · Acceptance letter]

21 Oct 2021

PONE-D-21-17352R2 

Mg, K-containing microparticle: a possible active principle of a culture extract produced by a microbial consortium 

Dear Dr. Higashinakagawa:

I'm pleased to inform you that your manuscript has been deemed suitable for publication in PLOS ONE. Congratulations! Your manuscript is now with our production department. 

Kind regards, 

on behalf of

Prof. Dr. Stefan Wölfl 

Academic Editor

PLOS ONE